# Identification and Structural Elucidation of Anti-Inflammatory Compounds from Chinese Olive (*Canarium Album* L.) Fruit Extracts

**DOI:** 10.3390/foods8100441

**Published:** 2019-09-26

**Authors:** Yueh-Hsiung Kuo, Yu-Te Yeh, Sih-Ying Pan, Shu-Chen Hsieh

**Affiliations:** 1Department of Chinese Pharmaceutical Sciences and Chinese Medicine Resources, China Medical University, Taichung 40402, Taiwan; kuoyh@mail.cmu.edu.tw; 2Department of Biotechnology, Asia University, Taichung 41354, Taiwan; 3Chinese Medicine Research Center, China Medical University, Taichung 40402, Taiwan; 4Departments of Medicine and Biological Chemistry, Johns Hopkins University School of Medicine and Johns Hopkins All Children’s Hospital, St. Petersburg, FL 33701, USA; d02641002@ntu.edu.tw; 5Institute of Food Science and Technology, National Taiwan University; Taipei 106, Taiwan; a10062001@hotmail.com

**Keywords:** amentoflavone, Chinese olive (*Canarium album* L.), protocatechuic acid, sitoindoside I, tetrahydroamentoflavone

## Abstract

Chinese olive (*Canarium album* L.), a rich source of polyphenols, can be used as a functional food ingredient. We previously showed that the ethyl acetate fraction of this extract (CO-EtOAc) is an effective anti-inflammatory agent. Therefore, here, we aimed to screen the bioactive fractions extracted from CO-EtOAc using different isolation techniques, and purify the bioactive compounds based on their cytotoxic and anti-inflammatory abilities. CO-EtOAc was fractionated using silica gel and Sephadex column chromatography, and the active compounds were isolated and purified by high-performance liquid chromatography (HPLC). The structures of the resulting compounds were identified using proton nuclear magnetic resonance (NMR) spectra. Activity-directed fractionation and purification were used to identify the following active compounds with anti-inflammatory effects using lipopolysaccharide (LPS)-stimulated mouse macrophages: sitoindoside I, amentoflavone, tetrahydroamentoflavone and protocatechuic acid. For the first time, sitoindoside I and tetrahydroamentoflavone were isolated from Chinese olive, and the anti-inflammatory compounds of CO-EtOAc were identified, suggesting its potential for used as a health food ingredient.

## 1. Introduction

A plethora of accumulating evidence has proven that inflammation plays a central role in the pathogenesis of various disorders such as cancer, diabetes, obesity, and cardiovascular complications. Controlling inflammation is, therefore, of major importance in treating chronic inflammation-associated illnesses [1]. Lipopolysaccharide (LPS) accounts for 3–8% of the dry weight of the cell envelop in Gram-negative bacteria, and is one of the most powerful activators of macrophages in the production of pro-inflammatory cytokines [2]. The RAW264.7 macrophage cell line is extensively used as a cell model in the study of the signal transduction pathway upon activation of pro-inflammatory mediators [3], including nuclear factor-kappaB (NF-κB), activator protein-1 (AP-1), interleukin-1 beta (IL-1β), IL-6, IL-8 and nitric oxide (NO) [4,5,6]. Of these mediators, NO serves as an important player in innate immunity due to its role in macrophage-mediated cytotoxicity for the killing of intracellular pathogens [7]. The wide distribution of macrophages in body tissues is not only critical to the function of the mammalian immune system; it is also the first line of host defense against foreign agents, prior to leukocyte migration, via the production of various pro-inflammatory mediators such as the short-lived free radical, NO [8,9]. Therefore, inhibiting NO production in LPS-stimulated RAW264.7 cells can be adopted to screen various anti-inflammatory drugs. The overproduction of NO under certain conditions can lead to a myriad of physiological and pathological processes, such as acute and chronic inflammation-induced diseases [10,11]. Therefore, more attention has now turned to the development of new drugs with a potency that will inhibit NO production; this production can be detected using a fast and simple colorimetric assay, the Griess reaction-based method [12,13].

Compared to animals, plants are considered a better source of natural antioxidant molecules including terpenoids, phenolic compounds, tannins, flavonoids, alkaloids, coumarins, and other metabolites. Considerable attention has been focused on the use of antioxidants, especially natural antioxidants, to inhibit NO production [14,15]. In Asia, the Chinese olive (*Canarium album* L.) fruit has been used as a folk herbal medicine to treat faucitis, stomatitis, hepatitis and toxicosis [16]. This is because of the fruit’s ability to exhibit a wide range of pharmacological effects, inclusive of anti-oxidation [17], anti-cancer [18], anti-lipid accumulation [19], and attenuated metabolic dysfunction [20]. Previously, we reported that LPS-induced inflammation was significantly suppressed by the addition of an ethyl acetate fraction of the Chinese olive fruit extract (CO-EtOAc) [21]. Our data also found CO-EtOAc could affect inducible nitric oxide synthase (iNOS) and cyclooxygenase-2 (COX-2), the two major downstream targets of NF-kB which is a key transcription factor in the triggering of inflammation signaling [21]. The bioactive compounds contributing to the CO-EtOAc-mediated anti-inflammatory effects have, however, remained poorly investigated. Herein, we aimed to identify the anti-inflammatory compounds within CO-EtOAc, using an NO production assay and the RAW264.7 macrophage-based platform.

## 2. Materials and Methods

### 2.1. General Experimental Procedures

Silica gel (63–200 mesh; Merck, Germany) and Sephadex^®^ LH-20 (25–100 μm) were the resins used for column chromatography (CC). The silica gel 60 F-254 (Merck, Darmstadt, Germany) resin was used for thin-layer chromatography (TLC), with the compounds visualized by spraying with 10% (*v*/*v*) H_2_SO_4_ in an ethanol solution and heating for 10 min at 95 °C. HPLC chromatograms were obtained using an LC-6A instrument and an IOTA-2 RI-detector (Shimadzu, Kyoto, Japan). A Phenomenex Luna silica column (5 μm, 10 × 250 mm, 3 mL·min^−1^ flow rate) was used for normal-phase separations. To elucidate the chemical structures, the isolated active compounds were identified by nuclear magnetic resonance (NMR). ^1^H NMR spectra were run on a Bruker Avance-500 MHz FT NMR (Bruker, Rheinstetten, Germany) with CDCl_3_ and tetramethylsilane (TMS) as the solvent and internal standard, respectively. A multiscan microplate reader (Thermo Fisher Scientific, Waltham, MA, USA) was used for the water-soluble tetrazolium salt (WST)-1 and NO assays.

### 2.2. Chemicals and Solvents

Sitoindoside I, amentoflavone, protocatechuic acid, LPS (endotoxin from *Escherichia coli*, serotype 0127:B8), Dulbecco’s Modified Eagle Medium (DMEM) culture medium, fetal bovine serum (FBS), antibiotics, and trypsin were purchased from Sigma-Aldrich (St. Louis, MO, USA). The WST-1 reagent was purchased from Abcam (Cambridge, MA, USA). All solvents were either American Chemical Society (ACS) or high-performance liquid chromatography (HPLC) grade and were obtained from JT Baker (Phillipsburg, NJ, USA).

### 2.3. Plant Material and Extraction Procedures

Chinese olive fruits (11 kg) were obtained from Baoshan Township (Hsinchu County, Taiwan). Figure 1 shows the scheme used to prepare and purify the effective compounds from the fruit. In brief, fresh Chinese olive fruits (11 kg) were crushed and extracted with water (60 L) for 2 h at room temperature (25 °C), and the resulting slurries centrifuged for 10 min at 1000 × *g* to separate the aqueous portion. The residual portion was dried (7.3 kg, using freeze dryer) and extracted with methanol (MeOH, 15 L); this process was repeated three times. The combined MeOH extract was evaporated under reduced pressure to obtain a crude extract (1786 g) which was suspended in H_2_O (1.8 L); the solution was then partitioned between H_2_O and EtOAc (1.8 L) three times. The EtOAc fraction (150 g) was poured onto a silica gel (3.0 kg) column using *n*-hexane–EtOAc and MeOH–EtOAc mixtures as solvent systems to collect 10 fractions (frs.): fr. A (1.1 g), fr. B (18.7 g), fr. C (19.9 g), fr. D (1.9 g), fr. E (2.3 g), fr. F (13.7 g), fr. G (11.3 g), fr. H (39.6 g), fr. I (24.3 g), and fr. J (8.6 g). Fr. F was separated by TLC using acetone/ dichloromethane (3:7, *v*/*v*) and observed under UV at 254nm (Appendix A). The fractions with similar components were pooled into the following 6 portions: fr. F1 (2.0 g), fr. F2 (1.45 g), fr. F3 (1.86 g), fr. F4 (2.27 g), fr. F5 (3.30 g), and fr. F6 (2.73 g). Fr. F5 (3.0 g) was further separated by Sephadex LH-20 column chromatography using dichloromethane/methanol (1:1, *v*/*v*) as the eluent, in an isocratic manner. Based on the TLC profile after spraying with 10% H_2_SO_4_, fractions that were similar were combined to obtain the following ten subfractions (Appendix A): fr. F5a (0.14 g), fr. F5b (0.32 g), fr. F5c (0.51 g), fr. F5d (1.16 g), fr. F5e (0.19 g), fr. F5f (0.03 g), fr. F5g (0.44 g), fr. F5h (0.03 g), fr. F5i (0.03 g), and fr. F5j (0.01 g). Finally, to elucidate the chemical structures, frs. F5b, F5e, and F5f were purified by HPLC and identified using ^1^H NMR spectroscopy. For storage, all the extraction was stocked at −20°C.

### 2.4. Cell Culture

RAW264.7, a mouse macrophage cell line, was obtained from the American Type Culture Collection (ATCC TIB71, Rockville, MD, USA). The RAW264.7 cells were cultured in DMEM supplemented with 10% heat-inactivated fetal bovine serum (FBS) and antibiotics (100 U/mL penicillin and 100 μg/mL streptomycin). The cells were continuously maintained in the exponential phase via frequent passage, (every 2–3 days) depending on the total cell number, at 37 °C in a humidified incubator supplied with 5% CO_2_.

### 2.5. Water-Soluble Tetrazolium Salt (WST)-1 Cell Proliferation Assay 

The WST-1 assay was developed as a colorimetric and non-radioactive assay which was performed to evaluate the proliferation and survival of viable cells. RAW264.7 cells were seeded in a 96-well plate and incubated with various concentrations of extracts or fractions from CO-EtOAc, for a period of one day prior to the WST-1 assay. At the end of treatment, the cells were processed for the assay according to the manufacturer’s instructions. Briefly, WST-1 reagents were added to the culture medium and incubated for 2 h at 37°C. Prior to measuring, the plate was placed on an orbital shaker for 1 min to ensure homogeneity of the samples, and the cell viability determined using an enzyme-linked immunosorbent assay (ELISA) plate reader; in this assay, the absorbance of the samples at 450 nm was measured, and the absorbance at 620 nm also measured as a background reference.

### 2.6. Nitric Oxide (NO) Assay

RAW264.7 cells were seeded in a 24-well plate and incubated with various concentrations of extracts or fractions from CO-EtOAc, in the presence or absence of LPS (1.0 μg/mL) for 24 h. The NO levels in the culture supernatants were assessed using the Griess method to measure nitrite (NO_2_^−^), a stable breakdown product of NO. Briefly, the culture supernatant was added into a 96-well plate, in duplicate, at a volume of 50 μL, followed by the addition of 100 μL Griess I (1% sulfanilamide in 5% phosphoric acid) and 100 μL Griess II (0.1% naphthylenediamine in 2.5% phosphoric acid) reagents. The plate was then incubated for 10 min at room temperature. The absorbance was read at 570 nm using an automated microtiter plate reader. Nitrite concentration was calculated by comparing the results to a standard curve generated with sequential dilutions of sodium nitrite.

### 2.7. Statistical Analysis

All results are normally distributed and expressed as mean ± SD. Differences among means were tested for statistical significance using one-way analysis of variance (ANOVA), followed by Tukey’s multiple test. All statistical analyses were carried out using the GraphPad 6.0 software (GraphPad Software Inc., La Jolla, CA, USA), with statistical significance set at *p* < 0.05.

## 3. Results

### 3.1. The Effects of CO-EtOAc on Cell Viability and Lipopolysaccharide (LPS)-Induced NO Production in RAW264.7 Cells

To examine the anti-inflammatory effect of CO-EtOAc, we opted to use the RAW264.7 cell model because of its quick response to inflammation signaling. As shown in Figure 2A, the unstimulated macrophages (control) produced NO (3.97 ± 0.31 μM), an indicator of inflammatory response, in the culture medium at background level. LPS induced an evident increase in NO production (23.63 ± 0.6 μM), indicating that the RAW264.7 cells were responsive to the inflammation-inducing reagent. To investigate the anti-inflammatory effect of CO-EtOAc, various concentrations (25, 50, 100, 200, 400, and 800 μg/mL) were co-treated with LPS (1 μg/mL) for 24 h. CO-EtOAc reduced the production of NO triggered by LPS, to 9.31 ± 2.10 μM, 5.2 ± 1.32 μM, 1.2 ± 0.10 μM, 0.4 ± 0.03 μM, 0.7 ± 0.05 μM and 0.25 ± 0.02 μM at concentrations of 25, 50, 100, 200, 400, and 800 μg/mL, respectively (Figure 2B). In addition, we observed that at doses above 100 μg/mL, CO-EtOAc exhibited cytotoxicity which might have affected the resulting NO production. However, at 25 and 50 μg/mL, CO-EtOAc exhibited no effect on cell survival (Figure 2A), and still presented strong anti-inflammatory capabilities in a dose-dependent manner (Figure 2B).

### 3.2. The Effects of the Subfractions from CO-EtOAc on Cell Viability and LPS-Induced NO Production in RAW264.7 Cells

To further identify the active compounds with anti-inflammatory activities in CO-EtOAc, we loaded CO-EtOAc (150 g) onto a silica gel column, using ethyl acetate-*n*-hexane and ethyl acetate-methanol gradients as the chromatography mobile phases to obtain 10 fractions. These factions were then tested for their cell viability and anti-inflammatory response using the WST-1 and Griess assays, respectively. As shown in Figure 3A, unlike the untreated sample, fractions (frs.) A, B, F, and G at 100 µg/mL had no cytotoxic effect following 24 h of incubation; instead, a significant increase in cell survival rate was observed compared to that with the parent fraction, CO-EtOAc. This suggested that the cytotoxic compounds had already separated into their respective fractions. Consistent with our speculation, the cytotoxicity in fr. D and fr. E was enhanced when compared to that with CO-EtOAc at the lower doses, 25 µg/mL and 50 µg/mL. To investigate the anti-inflammatory effects of the CO-EtOAc-derived fractions, we compared NO production abilities among the fractions obtained and the parent, CO-EtOAc. As displayed in Figure 3B, at high concentrations (50 µg/mL and 100 µg/mL), LPS-induced NO production was suppressed, without any significant differences observed among all fractions after 24 h of treatment. At 25 µg/mL, frs. D, E, F, and H remarkably decreased the NO level compared to CO-EtOAc at the same concentration. This indicated that the separation had enriched the bioactivity to inhibit inflammation. Since fr. F showed potent inhibitory effect against LPS-induced NO production and had no influence on cell viability, we selected fr. F for further separation and purification.

### 3.3. The Effects of the Fraction F Subfraction on Cell Viability and LPS-Induced NO Production in RAW264.7 Cells

After collecting fr. F using silica gel column chromatography, a combination was performed based on the similarity of patterns observed in TLC for each collection. We therefore collected six partially-purified fractions (fr. F1 to fr. F6) using the above-mentioned process. Figure 4A shows that all subfractions from fr. F, except fr. F5, inhibited cell growth compared to the parent, fr. F. However, a better inhibitory effect no longer existed at lower concentrations (10 μg/mL and 5 μg/mL) in the subfractions, when compared to fr. F. We then sought to determine the anti-inflammatory effects of the subfractions, fr. F1–F6. As shown in Figure 4B, no observable enrichment in the anti-inflammatory ability was observed in all subfractions; fr. F6 even exhibited a lower anti-inflammatory ability than the other fractions and the parent fraction. The above results aided in the selection of fr. F5 for further isolation and bio-guided assays. This is because of all the subfractions, fr. F5 still exhibited a comparable anti-inflammatory effect to its parent, fr. F. In addition, this subfraction showed no cytotoxicity when compared to the other subfractions from fr. F.

### 3.4. The Effects of the Subfractions from Fraction F5 on Cell Viability and LPS-Induced NO Production in RAW264.7 Cells

To separate the compounds exhibiting different chemical properties within the fr. F5 subfraction, we used size exclusion Sephadex LH-20 column chromatography to elute 76 glass bottles of this fraction using dichloromethane/methanol (*v*/*v* = 1:1). Thus, a total of 76 fractions was collected and combined using the results of TLC to generate ten subfractions. We observed that the following frs, F5g, F5h, F5i, and F5j, exhibited a much more significant inhibitory effect on cell viability than the parent, fr. F5, at all concentrations tested (25, 10 and 5 µg/mL) (Figure 5A). We therefore measured the anti-inflammatory effects of subfractions F5a to F5f as they were deemed non-toxic to the RAW264.7cells. As shown in Figure 5B, frs. F5b, F5e, and F5f showed potent inhibitory activity toward NO production at 5 µg/mL unlike the parent, fr. F5.

### 3.5. Identifying the Chromatographic Peaks in frs. F5b, F5e, and F5f 

To identify the active compounds within frs. F5b, F5e and F5f, which were the most effective subfractions at all concentrations investigated (Figure 5B), we performed separation via preparative HPLC. The mobile phase for fr. F5b was a mixture of 30% acetone/dichloromethane, and the flow rate was 3 mL/min. The fraction retention time at 12.9, 17.2 and 18.7 min was collected and further analyzed by NMR. For frs. F5e and F5f, the mobile phase was a mixture of 45% ethyl acetate/dichloromethane, and the flow rate was 3 mL/min. The fraction retention time for fr. F5e at 7.3, 9.3 and 12.5 min and the fraction retention time of fr. F5f at 8.6 min were collected and further analyzed by NMR; Appendix A for frs. F5e and F5f, respectively. Based on the ^1^H NMR spectra (Figure 6), compound **1** (*R*_t_ =18.7 min) from fr. F5b was identified as sitoindoside I, C_51_H_90_O_7_; compound **2** (*R*_t_ = 7.3 min) from fr. F5e was identified as tetrahydroamentoflavone (THA), C_30_H_22_O_10_; compound **3** (*R*_t_ = 12.5 min) from fr. F5e was identified as amentoflavone, C_30_H_18_O_10_; and compound **4** (*R*_t_ = 8.6 min) from fr. F5f was identified as protocatechuic acid, C_7_H_6_O_4_ (Appendix A).

### 3.6. The Effects of Sitoindoside I, Amentoflavone, and Protocatechuic Acid on Cell Viability and LPS-Induced NO Production in RAW264.7 Cells

To further confirm the anti-inflammatory effects of sitoindoside I, amentoflavone and protocatechuic acid, WST-1 and NO assays were conducted using these compounds. As presented in Figure 7A, the three compounds had no effect on cell viability at the indicated concentrations (10 μg/mL and 5 μg/mL). We then sought to determine the anti-inflammatory effects of the three compounds. The results showed that sitoindoside I, amentoflavone, and protocatechuic acid at concentrations of 10 μg/mL and 5 μg/mL significantly inhibited LPS-stimulated NO production compared to LPS-only treatment (Figure 7B).

## 4. Discussion

Extensive efforts have been dedicated to identifying bioactive compounds in herbal plants; this is due to overwhelming evidence from epidemiological, in vivo, in vitro, and clinical trial data, indicating that a plant-based diet can improve human health by reducing the risk of chronic diseases. Polyphenols are largely found in plants and are characterized by the presence of multiple phenol structural units categorized into many classes such as phenolic acids, flavonoids, lignans, and stilbenes. High dietary intake of polyphenols results in protective effects being exerted against inflammatory diseases [22]. Recently, phytosterols were claimed to provide protection against different types of inflammatory diseases [23,24]. β-sitosterol, campesterol and stigmasterol are predominant phytosterols in human herbal nutrition, accounting for 65%, 30% and 3% of dietary contents, respectively [25]. Rapid onset and development of many diseases often result in inflammation and extremely high mortality rates, [26]. We thus believe that suppressing NO production, as performed by the Chinese olive extract, may satisfy the unmet need of being able to control the inflammatory process. In the present study, we used an inflammation responsive cell platform to direct the separation of fractions, from the Chinese olive extract with anti-inflammatory activities. By performing this separation, we have successfully identified, for the first time, many compounds in Chinese olive. Sitoindoside I (compound **1**), a beta-sitosterol-related derivative with a glucose and a palmitate linked to C3 and C1 of a beta-sitosterol ring, is involved in the curative properties exhibited by plants in improving inflammation, viral damage, ulcer, cancer development, as well as the enhancement of the immune system [27,28]. For the first time, sitoindoside I has been identified in Chinese olive. The importance of sitoindoside I in immune modulation has been proven due to the mixture of β-sitosterol and sitoindoside I proving effective in modulating the behavior of T-helper cells [29]. In addition, administering sitoindoside I as a single therapy was reported to result in a better anti-inflammatory effect than the mixture of β-sitosteryl glucoside and stigmasterol. It is suggested that the moiety of palmitic acid and glucose contributes to the enhanced anti-inflammatory activity of sitoindoside I, when compared to that of its backbone structure, sitosterol [30]. In the current study, we isolated two biflavonoids, amentoflavone (compound **3**) and THA (compound **2**). The diversity in biflavonoids is a result of many different substitution positions, which generates a large number of natural derivatives. Generally, the dimeric form of flavonoids (homo or hetero) connected with a C-O-C or C-C bond at diverse positions, illustrate the characteristic structure of biflavonoids [31]. Even though numerous biflavonoids have been isolated and structurally described to this day, medicinal plants that contain biflavonoids as a major constituent, are not widely distributed [32]. Amentoflavone, a common biflavonoid existing globally in a number of medicinal plants, is considered a dimer of two apigenins, and has a covalent C3′-C8” linkage and six hydroxyl groups at C5, C7, C4′, C5”, C7”, and C4”’ [33]. Amentoflavone, another chemical constituent identified in Chinese olive [34], has attracted numerous research attention due to its exciting pharmacological potential. Its bioactivities have been extensively studied, especially its regulatory effect on oxidant/antioxidant balance in inhibiting the production of inflammatory mediators [35,36,37]. Amentoflavone can also inhibit the ability of NF-κB-mediated iNOS to reduce the production of NO, thereby preventing an inflammatory response [38]. The hydroxyl groups in amentoflavone are also easily hydrogenated; thus, the hydrogenated product, THA, identified in this present study, is a derivative of amentoflavone. To the best of our knowledge, there remain very few reports detailing the biological activities of THA. Therefore, the present study is the first to report its existence and activity in Chinese olive. THA has, however, been reported as a potent inhibitor of xanthine oxidase [39], which is a major enzymatic source in the generation of reactive oxygen species, and plays fundamental roles in inflammation [40]. Another study also revealed that THA exhibited anti-inflammatory activity in vivo and cyclooxygenase inhibitory activity in vitro [41]. Besides the above compounds, we also identified protocatechuic acid (PCA, compound **4**) from the Chinese olive fruits. PCA is a widely distributed phenolic acid and is present in most of the edible plants used in folk medicine. PCA has structural similarities to gallic acid, caffeic acid, vanillic acid, and syringic acid [42]. With the development of modern pharmacology, more evidence has proven the bioactivities of PCA, including its anticancer, antiulcer, antiaging, antifibrotic, and therapeutic effects in neurology [43]. PCA suppressed the production of the proinflammatory cytokines, TNF-alpha and IL-1 beta, the inflammatory mediators, NO and prostaglandin E2 (PGE2), and the gene expression of iNOS and COX-2, in RAW264.7 cells [44]. Comparing the activity and relationship, we thought that more hydroxyl groups would lead to stronger anti-inflammatory activity as amentoflavone (6 hydroxyl groups) exhibited higher activity than protocatechuic acid (2 hydroxyl groups) and stioindoside I (3 hydroxyl groups). The phenomenon derives from that fact that after proton releasing, the anion of hydroxyl group becomes a good electron donor to stabilize other free radicals while converting itself to radical cation, which could be stabilized by its resonance structure [45]. In addition, protocatechuic acid only obtains two hydroxyl groups, and the strong electron-withdrawing activity of carboxyl group at para position of hydroxyl group will weaken the oxidation activity of the hydroxyl group, for it would dampen the resonance structure of phenol, thus decreasing the anti-inflammation activity. These findings suggest that the active compounds (1–4) may exert their anti-inflammatory effect by inhibiting the production of NO.

## 5. Conclusions

In this study, we demonstrated the anti-inflammatory effects of the ethyl acetate fraction of the Chinese olive fruits extract. Furthermore, we identified sitoindoside I, THA, amentoflavone, and protocatechuic acid as the compounds responsible for the anti-inflammatory activity exhibited by the fruit extract. Of the 4 compounds, this study is the first to identify sitoindoside I and THA in Chinese olive. Importantly, the significance of these compounds in managing inflammation related to pathological conditions is worthy of further investigations. This study has provided additional scientific support for the use of Chinese olive fruits in Traditional Chinese Medicine to treat inflammation-associated diseases, and to prove its potential for development as a natural nutraceutical and functional food ingredient.

## Figures and Tables

**Figure 1 foods-08-00441-f001:**
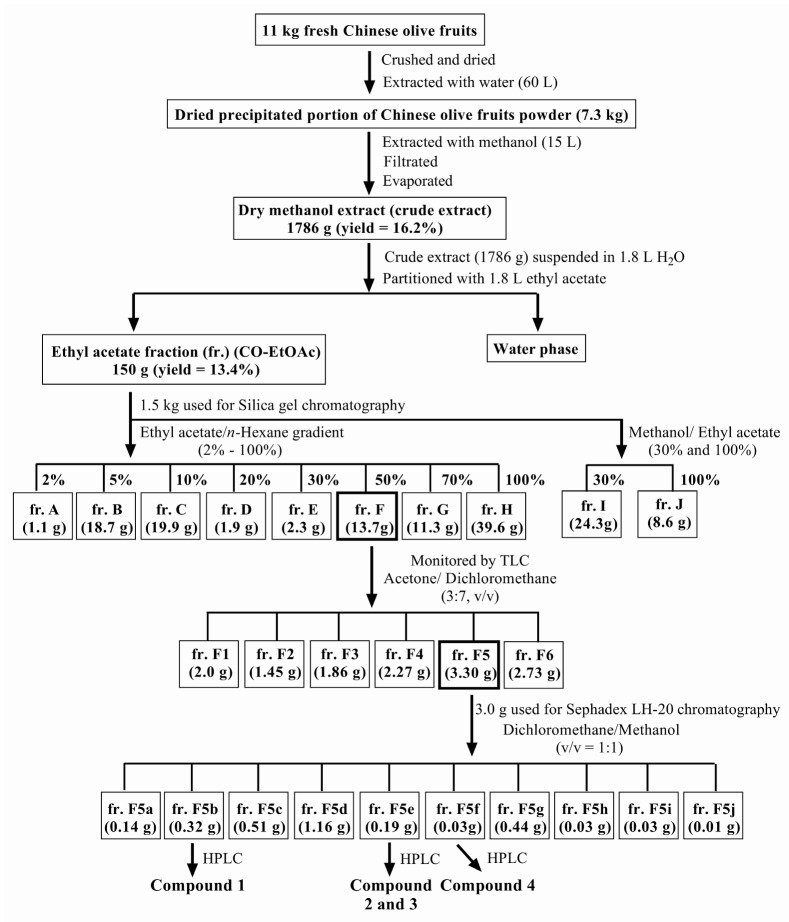
Scheme used to prepare and isolate the anti-inflammatory fractions and compounds in the Chinese olive fruits.

**Figure 2 foods-08-00441-f002:**
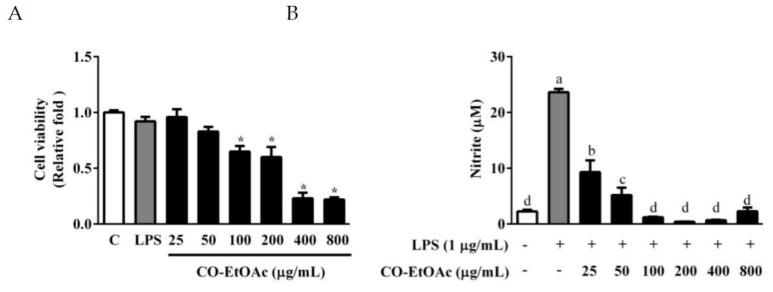
The effects of different concentrations of CO-EtOAc on cell viability (**A**) and nitric oxide (NO) production (**B**) in RAW264.7 cells. RAW264.7 cells were treated with various concentrations of CO-EtOAc (25, 50, 100, 200, 400, and 800 μg/mL) in the presence or absence of lipopolysaccharide (LPS) (1 μg/mL) for 24 h. (**A**) Cell viability was measured using the WST-1 assay, and the (**B**) NO production in the cell culture supernatants determined using the Griess assay. The results are expressed as means ± standard deviation (SD) from three independent experiments. For cell viability, * indicates *p* < 0.05 when compared to the dimethyl sulfoxide (DMSO) control group. For NO production: values with different letters indicate significant difference (*p* < 0.05).

**Figure 3 foods-08-00441-f003:**
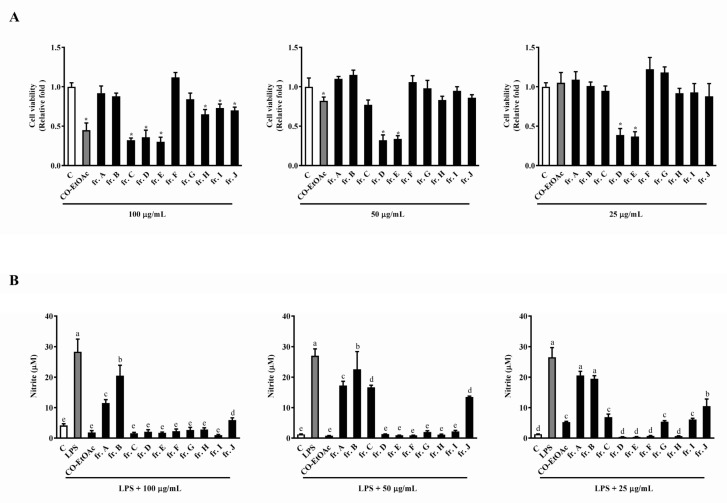
The effects of the subfractions from CO-EtOAc (at different concentrations) on cell viability and NO production in RAW264.7 cells. RAW264.7 cells were treated with various concentrations of CO-EtOAc (25, 50, and 100 μg/mL) in the presence or absence of LPS (1 μg/mL) for 24 h. (**A**) Cell viability was measured using the WST-1 assay, and the (**B**) NO production in the cell culture supernatants determined using the Griess assay. The results are expressed as means ± SD from three independent experiments. For cell viability: * indicates *p* < 0.05 when compared to the DMSO control group. For NO production: values with different letters indicate significant difference (*p* < 0.05).

**Figure 4 foods-08-00441-f004:**
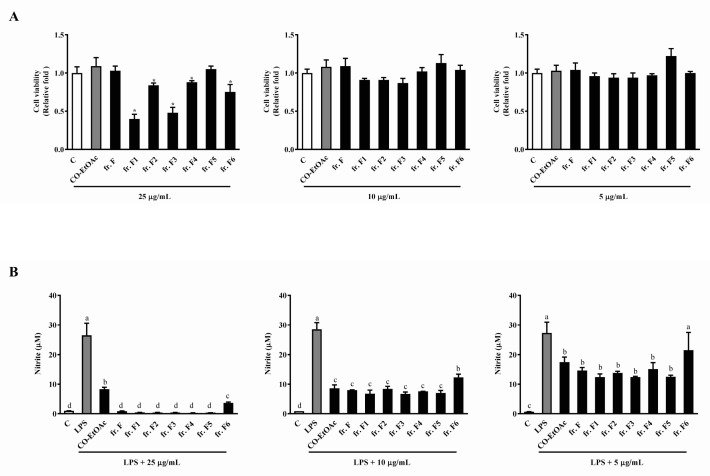
The effects of the subfractions from fr. F (with different concentrations) on cell viability and NO production in RAW264.7 cells. RAW264.7 cells were treated with various concentrations of CO-EtOAc (5, 10, and 25 μg/mL) in the presence or absence of LPS (1 μg/mL) for 24 h. (**A**) Cell viability was measured using the WST-1 assay, and the (**B**) NO production in cell culture supernatants determined using the Griess assay. The results are expressed as means ± SD from three independent experiments. For cell viability: * indicates *p* < 0.05 when compared to the DMSO control group. For NO production: values with different letters indicate significant difference (*p* < 0.05).

**Figure 5 foods-08-00441-f005:**
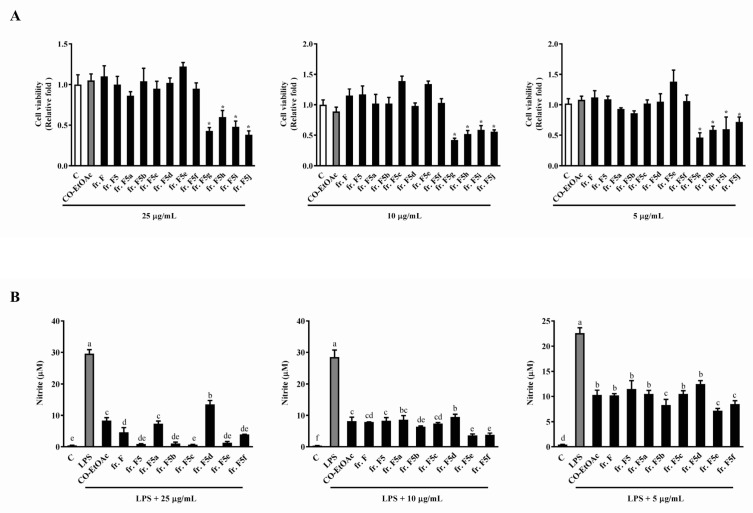
The effects of subfractions from fr. F5 (at different concentrations) on cell viability and NO production in RAW264.7 cells. RAW264.7 cells were treated with various concentrations of CO-EtOAc (5, 10, and 25 μg/mL) in the presence or absence of LPS (1 μg/mL) for 24 h. (**A**) Cell viability was measured using the WST-1 assay, and the (**B**) NO production in the cell culture supernatants determined using the Griess assay. The results are expressed as means ± SD from three independent experiments. For cell viability: * indicates *p* < 0.05 when compared to the DMSO control group. For NO production: values with different letters indicate significant difference (*p* < 0.05).

**Figure 6 foods-08-00441-f006:**
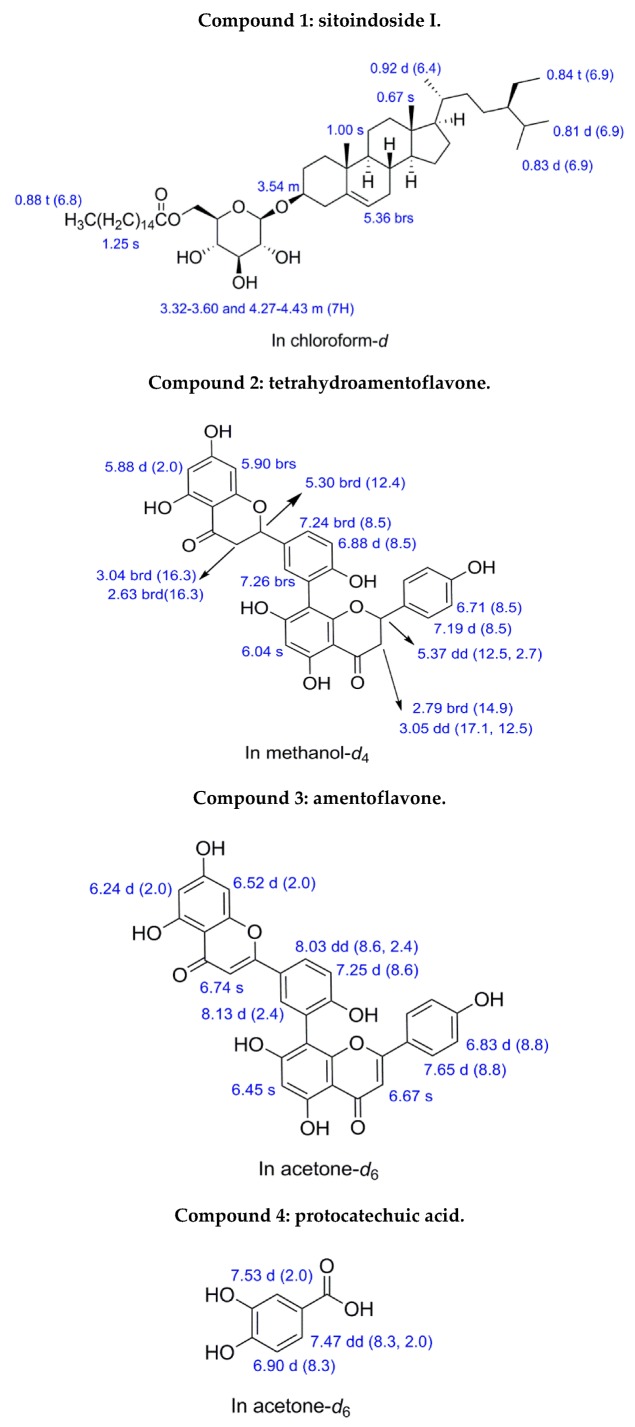
H^1^-NMR (nuclear magnetic resonance) data of sitoindoside I, tetrahydroamentoflavone, amentoflavone, and protocatechuic acid (500 MHz, acetone-d_6_).

**Figure 7 foods-08-00441-f007:**
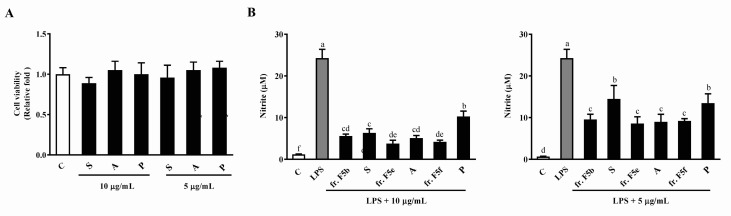
The effects of sitoindoside I, amentoflavone, and protocatechuic acid on cell viability and NO production in RAW264.7 cells. RAW264.7 cells were treated with various concentrations of frs. F5b, F5e, and F5f, sitoindoside I (S), amentoflavone (A), and protocatechuic acid (P) (10 and 5 μg/mL) in the presence or absence of LPS (1 μg/mL) for 24 h. (**A**) Cell viability was measured using the WST-1 assay, and the (**B**) NO production in cell culture supernatants determined using the Griess assay. The results are expressed as means ± SD from three independent experiments. For NO production: values with different letters indicate significant difference (*p* < 0.05).

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
