# Peer review of "Identification and Structural Elucidation of Anti-Inflammatory Compounds from Chinese Olive (Canarium Album L.) Fruit Extracts"

_foods, 2019, doi:10.3390/foods8100441_

Round 1

Reviewer 1 Report

The manuscript addresses the isolation and identification of some bioactive compounds from Chinese olive (Canarium album L.) fruit, together with the evaluation of their cytotoxic and anti-inflammatory effects.

- Line 52: Please change [12] [13] with [12,13]. - Section 2.3. 1) Please explain how the sample was dried (line 90). 2) Please explain why the sample was stored at 4°C and not t –20°C, as it is usual. 3) Please also explain the choice of water as solvent for the extraction, altough it is known that a mixture of water /ethanol is a better solvent for the extraction of phenolic compounds (see also reference 17 of your manuscript). 5) Specify the weight of the sample extracted, and also the amount of the solvent used in the first step. - The Italic font should be removed in several lines (e.g. line 184, line 201, etc) - Lines 244-246: please change Compound 2 with compound 2 (remove the capital letter); please do the same also for compounds 3 and 4. - Figure 6: the structures are in different format. - Line 297: Please remove the comma after “therapy”.

Author Response

Dear Professor

Thank you very much for considering our manuscript entitled" Identification and structural elucidation of anti-inflammatory compounds from Chinese olive (Canarium album L.) fruit extracts " foods-598377 to be published in foods. We appreciate the comments from you and have listed our point-to-point response in the followings. The changes we made in the manuscript have been highlighted in red. Thank you again for giving us the chance to revise our article.

Sincerely yours

Shu-Chen

Shu-Chen Hsieh, Ph. D.

Professor

Institute of Food Science and Technology

National Taiwan University

No. 1, Sec. 4, Roosevelt Road, Taipei, 10617, Taiwan

Office: +886-(0)-2-33669871

Fax: +886-(0)-2-2362-0849

Reviewer 2 Report

In this study, authors reported activity-guided separation and identification of anti-inflammatory compounds in Chinese olive. Experiments were performed logically and manuscript clearly described the importance of this research. However, some points need to be revised.

Line 4: “Canarium album” à italics

Line 60-61: Which references are for the previous reports on anti-inflammation of Chinese olive? Need to check and add pertinent citation.

Line 116: What is WST-1 assay? Need to amend explanation. i.e. Cell proliferation assay…

Line 87-108: NMR procedures for structure identification should be added including NMR spectrums, pulse sequences applied etc.

Line 235-247: What were the compounds in Fr.5b at R.T. 12.9 and 17.2 min?

What were the compounds in Fr.5e at R.T. 9.3 and 12.5 min?

Need to clarify the relationship of these compounds and the activity.

References need to be revised according to the citation format.

Author Response

(The authors gave the same response as above.)
